# Methods for Assessing Neurodevelopmental Disorders in Mice: A Critical Review of Behavioral Tests and Methodological Considerations Searching to Improve Reliability

**DOI:** 10.3390/neurosci6020027

**Published:** 2025-03-27

**Authors:** Boniface Echefu, Maria Becker, Dan Stein, Asher Ornoy

**Affiliations:** 1Department of Morphological Sciences and Teratology, Adelson School of Medicine, Ariel University, Ariel 40700, Israel; bonifacee@ariel.ac.il (B.E.); mariabe@ariel.ac.il (M.B.); dansh@ariel.ac.il (D.S.); 2Jerusalem Multidisciplinary College, Jerusalem, Israel; 3Hebrew University Hadassah Medical School, Jerusalem 9112102, Israel

**Keywords:** neurobehavioral disorders, behavioral tests, mice, pitfalls, scoring system

## Abstract

Many neurobehavioral tests are used for the assessment of human-like behaviors in animals. Most of them were developed in rodents and are used for the assessment of animal models that mimic human neurodevelopmental and neuropsychiatric disorders (NDDs). We have described tests for assessing social behavior, social interaction, and social communication; tests for restricted and repetitive behaviors; tests for cognitive impairment, for sensory stimuli, for anxiety like behavior, and for motor coordination deviations. These tests are used to demonstrate autistic-like behavior as well as other NDDs. We described possible general pitfalls in the performance of such studies, as well as probable individual errors for each group of tests assessing specific behavior. The mentioned pitfalls may induce crucial errors in the interpretation of the results, minimizing the reliability of specific models of defined human NDD. It is imperative to minimize these pitfalls and use sufficient and reliable tests that can demonstrate as many of the traits of the human disorder, grade the severity of the specific deviations and the severity of the tested NDD by using a scoring system. Due to possible gender differences in the clinical presentations of NDD, it is important to carry out studies on males and females.

## 1. Introduction

Neurodevelopmental and neurobehavioral disorders (NDDs) such as attention deficit hyperactivity disorders (ADHDs), obsessive compulsive disorders (OCDs), and autism spectrum disorders (ASDs) are highly prevalent in humans [1]. Other conditions, including intellectual disability, bipolar disorder, Angelman syndrome, Tourette syndrome (TS) or schizophrenia (ZF), are associated with disruptions in the intricate sequence of brain developmental processes. These processes include proliferation, differentiation, cellular migration, synapse formation, and abnormalities in neural network connectivity [2,3,4]. The etiology and pathophysiology of NDDs are generally attributed to a combination of biological, genetic, epigenetic, and environmental factors [5,6].

Laboratory animals, particularly rodent models, remain crucial tools for face validation and predictive treatment developments for human NDD as well as other diseases. Studies employing rodents to investigate behavioral changes analogous to symptoms inherent in human NDDs often rely on several standardized neurobehavioral and neurological test paradigms. For several decades, these tests have been extensively utilized to understand behavioral phenotypes in mammals, contributing significantly to advancements in treatment development for human disorders, including NDDs. Nevertheless, these paradigms are not without challenges, limitations, and drawbacks. A major issue is the frequent failure to replicate results, which can often be attributed to variations across laboratories, procedural modifications, inconsistencies in data collection and presentation, litter effects, and the technical expertise of personnel, which collectively pose significant challenges to the reliability and generalizability of findings. Efforts to improve standardization, enhance reproducibility, and account for these variables remain critical to advancing the translational relevance of rodent models in NDD research. Many neurodevelopmental and neurobehavioral disorders in human are diagnosed using validated developmental and behavioral tools [1]. The American Psychiatric Association emphasizes the use of a standardized scoring system and the demonstration of characteristic behavioral changes that reflect specific disorders. Typically, diagnosis involves the use of well-validated questionnaires and behavioral tests complemented by a detailed history from the individual or caregivers. The behavior of the individual is assessed across multiple domains and systematically scored to ensure accuracy and reliability in diagnosis. Similarly, in animal models of neurodevelopmental disorders, achieving a proper and specific diagnosis requires validating a set of observable behavioral deviations and classifying the degree of their severity. This process involves the development of robust and reproducible scoring systems that align with established human diagnostic criteria, where applicable. Such classification is essential not only for understanding behavioral phenotypes in animal models but also for ensuring their translational relevance in preclinical research [5].

One of the major challenges in diagnosing NDD is the lack of specific biological markers for these disorders. This limitation extends to animal models, where researchers must primarily rely on behavioral tests that depict specific and typical behavioral changes. Moreover, even when observing behavioral traits that closely mimic human normal or abnormal behaviors, grading the severity of these changes remains challenging, as it often falls short of the precision required for human diagnosis.

An additional problem lies in the fact that behavior is influenced by many factors that are not necessarily related to the typical traits associated with the specific NDD. In addition to gender and age, factors such as thirst, hunger, sleep, noise, environmental and body temperature, duration of the test, and several others may all affect the performance on the tests and interfere with proper assessment of the results. Thus, these possibly neglected factors may add difficulties to the objective interpretation of the data.

In this review, we will discuss the most reliable and widely used tests in mouse models for identifying behavioral deviations that resemble those observed in various NDDs. We will especially highlight the potential pitfalls and limitations in the evaluation and interpretation of these test results. Addressing these challenges is critical for enhancing the validity and reliability of animal models data, ultimately improving the translational value of such studies for human clinical applications.

## 2. Possible Pitfalls When Using Animal Models of NDD: Potential General Problems

### 2.1. Litter Effect

Litter effects significantly impact embryonic and early postnatal studies due to the near impossibility of timing conception accurately [7]. Although awareness of litter effects remains relatively low, its effects are responsible for 30–60% of the variability commonly encountered in behavioral studies that tend to persist into adult life [8]. Studies focusing on early developmental stages should prioritize controlling for these effects to avoid misleading conclusions from small litter numbers or disproportionate litter size [7]. Despite the existence of literature highlighting the importance of litter control, many researchers preferred resource conservation practices using smaller underpowered samples. These practices end up leading to increased false positive rates and introduce artefacts that can obscure genuine treatment or genetic effects in test animals [9,10]. To address these issues, researchers should not only increase the number of litters to achieve higher statistical power but also equalize litter sizes so that all litters have uniform numbers starting on postnatal day (PND) one. Established strategies are suggested for addressing the litter effect, including, for example, selecting only one animal per litter at random, therefore using multiple animals per litter and averaging their values. Another possibility is applying a mixed-effects model for statistical analysis when using multiple animals per litter. For example, in the valproic acid (VPA) model of ASD, administering VPA to pregnant females induces ASD-like phenotypes in their offspring. In this type of experimental design, the sample size for a given treatment group should be determined by the number of litters (experimental unit) and not the total number of offspring. Using data from individual pups can lead to inaccurate results due to an inflated number of independent observations, which may increase the risk of false positives. Instead, average values from individual neonates can be utilized to enhance the precision of the litter mean, although they do not serve as separate experimental units. However, if the treatment occurs after birth, then the pups’ data must be handled individually. In the situation where one animal per litter is the experimental unit per litter, considerations should be given to the variation in weight among pups per litter. This is crucial considering that animals from a large litter tend to have extreme differences in some characteristics like relative weight and crown to rump length. Also, in the event of a significant difference in mating time or inaccurate recording of birth time, some litters in the same group may appear to be older than their age. In these cases, properly randomized selection of a single representative pup per litter is strongly encouraged. While employing weighted statistical analysis may be advantageous for evaluating neonatal results, it is often useful to seek professional advice [7,8,9,10]. It is encouraged to implement parallel methods of testing animals from different groups at relatively the same time instead of testing different treatment groups in isolation. In this manner, the handling of animals and variable experimental conditions at every point in time and space will be applicable to all the treatment groups. Failing to account for litter effects in this way may result in a significantly biased and severely underpowered outcome. Implementing appropriate measures to mitigate this significant source of variability is crucial for improving the rigor, accuracy, and reproducibility of experimental results, particularly in studies assessing NDDs. Proper control of litter effects will not only enhance the reliability of rodent model research but also strengthen the translational value of such studies for human clinical applications.

### 2.2. Uniformity of Pregnancy Timing

The importance of timing accuracy for all procedures including mating, vaginal plug observation, and treatment and parturition up to neurobehavioral testing sessions in studies of NDDs cannot be overemphasized. Considering the dramatic changes during the short estrous cycle and rapid rate of development in rodents’ central nervous system (CNS) in utero, mating hours and subsequent confirmation of copulation should be precise and strictly adhered to. Accurate timekeeping for all animals and treatment groups must be maintained for each and every procedure. This means that animals must be put up for mating at the same time of the evening, the vaginal plug must be observed as early as possible, and treatment must be administered at the same hour of the day across animals and across groups. Proper application of the Whitten and Lee-boot effect in study design is strongly encouraged to aid with synchronizing and maintaining reproductive events. That notwithstanding, the male-to-female mating ratio should not exceed 1:3 [11,12]. In studies that would involve administration/exposure of dams to chemical/biological manipulations, it is fundamental to ensure that all dams undergo a similar procedure (including handling) devoid of time-gap and stress [13,14]. Little time disparity could mean significant changes in studies that model environmental components of disease such as the VPA model of ASD. This is true in terms of the dramatic changes that happen per unit of time in the very short embryonic life of mice [15]. Accurate time keeping is also of utmost importance in performing behavioral testing for animal models. Although rodents are nocturnal animals, many researchers perform behavioral assays during the day. Consequently, some studies have been conducted at night/in darkness in attempts to understand the significance of the circadian cycle on behavior and how it affects results of studies conducted in the active and resting phases in rodents [16], as that could be crucial for behavioral tests performance. Regardless of the phases of rodent activity (day/night), we opine that conducting tests at the same time of the day, having a minimal number of litters (at most two) or few animals (at most 20) per day, and equal resting and inter-test intervals for all animals/litters are crucial to circumventing challenges related to time optimization in rodent behavioral test. This is evidently plausible as behaviors in rodents are traditionally measured in time units.

### 2.3. Housing and Testing Conditions

It is critical to briefly lay out some of the factors that could interfere with study outcomes of NDDs leading to conflicting results that are usually encountered between laboratories and experimental cohorts. Housing and testing conditions, as basic as they appear in animal studies, play a significant role in the phenotypes of rodent models of NDDs [17]. Studies in uncompartmentalized pre-weaning rodents showed that the males tend to be more social, compared to males separated early from the females, and that the estrous phase of females tends to influence their social behavioral outcomes and other sensory-motor reflexes [18,19]. It is recommended, therefore, to employ the Lee-Boot effect where applicable for the maintenance of the steady phase of estrus during behavioral studies especially for ASD models. Studies have reported that mice exhibit defensive behaviors when housed in proximity to rats. Therefore, it is advisable to avoid housing these species as neighbors. Additionally, when common equipment is used for both species, thorough cleaning is essential to minimize stress and cross-species influences [20,21]. Lighting conditions, temperature, and humidity must be tightly controlled and well documented for dams and their offspring at all times [22,23]. A strong strange smell or sound can be sources of stress for both dam and offspring and might impact the results of several types of neurobehavioral assays in high magnitude [13,14]. Cage and bedding replacement should be avoided at least two days prior to behavioral testing. Reports have emerged that rodents display several hours of increased anxiety, motor activity, and decreased sleep time on the days that the bedding materials and cages were changed [19,24,25,26]. Familiarization with and consistency of the experimenter in terms of scent, handling, and treatment techniques play a vital role in maintaining evenness of behavior in rodent models. When selecting a control group, it is important to recognize that the behavior of vehicle-control animals may differ from that of normal controls. This is because the vehicle itself can sometimes alter specific behavioral phenotypes, potentially affecting the study’s results [20]. While considering that some of the factors highlighted so far can be significantly influenced by genetic components of the test animals, it is also important to note that the test order, test number, and the intervals between tests might also be sources of conflicting results in neurobehavioral studies.

### 2.4. Phenotyping Specificity of Test Paradigms

Specificity of the test paradigm relative to different behavioral phenotype poses a complex burden on choosing a suitable test battery for studying a given NDD such as ASD. Similarly, different tests are currently available for several behavioral symptoms that characterize a specifically known disorder [17]. Regardless of the fundamental importance of these issues, less effort has been made to resolve them, leading to scarcity and paucity in literature. Ornoy et al. have pioneered the movement calling for a scoring system for ASD studies in an attempt to solve the problem of disease–model misidentification [5]. The study’s scope heralds the need for the implementation of a comprehensive scoring system for accurate face and construct validity of animal models and recapitulation of human ASD features as a prerequisite for the translation of preclinical findings into suitable clinical tools. Another compounding challenge is the contrasting outcome obtained when comparing behavioral test results from infancy, adolescence, and the adult stage of rodents for NDDs [15]. Life-stage-specific disparity could be minimized by starting behavioral tests early during infancy and progressing through adolescence to adult life, so that vital data that are developmental-stage specific are well captured [27].

Attempts to broaden the scope shows that a handful of studies are increasingly reporting on the commonality of phenotypic dysfunction and mechanisms shared among NDDs, especially SZ, ASD, and ADHD [17,28,29,30,31]. Considering these reports, the use of only a few tests to characterize a model of NDD will not yield a plausible conclusion on the specificity of that model. We believe that a model can be termed broadly as a neurodevelopmental model or more specifically as model of asocial/repetitive/communication behavior without necessarily ascribing it to a particular named NDD (e.g., SZ and OCD). This popular practice of hasty mischaracterization of rodent models of NDD is a major source of conflicting results in the field of NDD research. Such practices negatively impact the integrity of findings and hamper the translation of animal findings to human clinical use. Applying the proposed scoring system would help to identify not only ASD models but also other NDDs models for the evaluation and improvement of preventive and therapeutic interventions [5]. Regardless of the heterogenous features of NDDs, high co-occurrence, and the arbitrary nature of the boundaries that define individual disorders, having a comprehensive study design and robustized test battery remain the crucial ingredients for obtaining reliable and reproduceable neurobehavioral assay results.

## 3. Behavioral Tests Assessing Specific Behavioral Traits

### 3.1. Tests Assessing Social Behaviors 

#### 3.1.1. The Three Chamber Test for Sociability and Novel Social Preference (Figure 1)

In order to assess socialization and social preference in preclinical animal investigations, the three chamber (3-chamber) test is unarguably the most globally preferred behavioral paradigm as adopted from Crawley et al. [5]. The test is classically employed to test the sociability index in rodent studies. In addition, the test has been upgraded for the concomitant observation of preference in social novelty. Basically, sociability is defined in terms of the time a given subject spends with an age- and sex-matched conspecific. The apparatus (arena) consisted of a rectangular box partitioned into three separate chambers. The two outer chambers house two circular wire cages and are opened to the middle chamber via two square doors. The color and size of the apparatus and cages vary. Despite consistent principles and basic design, significant methodological variations exist across research laboratories, including differences in size, cage materials, test duration, room acclimatization periods, and behavioral outcome interpretation criteria.

In the test’s original version, subjects are introduced into the arena, usually into the center of the middle chamber, for habituation in the presence of the circular empty cages at opposite corners of the outer chambers; 5 to 10 min may be allocated for this stage, though it is often skipped entirely. In the second phase, a conspecific is placed in one of the circular cages, and the subject is allowed to explore the arena once again for 10 min. The objective of these two phases is to assess the sociability index for the subject [32,33,34]. The modified version is aimed at observing social preference/memory, which is achieved by incorporating a third phase is to the test, during which a second conspecific is placed in the remaining empty circular cage. Once again, the test subject is allowed 10 min to explore the arena with the first and second conspecific in the two circular cages. The time spent in each chamber, the time spent around each cage, and the total distance moved by the test subject are measured [35]. According to the principle of the test, animals that spend equal or more time with the empty cage than with the first conspecific in the second phase are considered to have deficits in social interaction. Similarly, subjects that spend more time with the first familiar conspecific than with the second conspecific in the modified version are judged to have impaired social memory and lack interest in social novelty [36,37].
Figure 1Behavioral tests for assessing social communication. The diagrams present a graphical illustration of mostly used behavioral tests for the evaluation of social behaviors. The three chamber test, reciprocal social interaction test, partition test, social conditioned place preference, and maternal isolation test as well as USV recordings are used for the assessment of paradigms of social interest and social memory; scent marking and olfactory discrimination tests are used for discrimination of social cues based on odor olfaction.
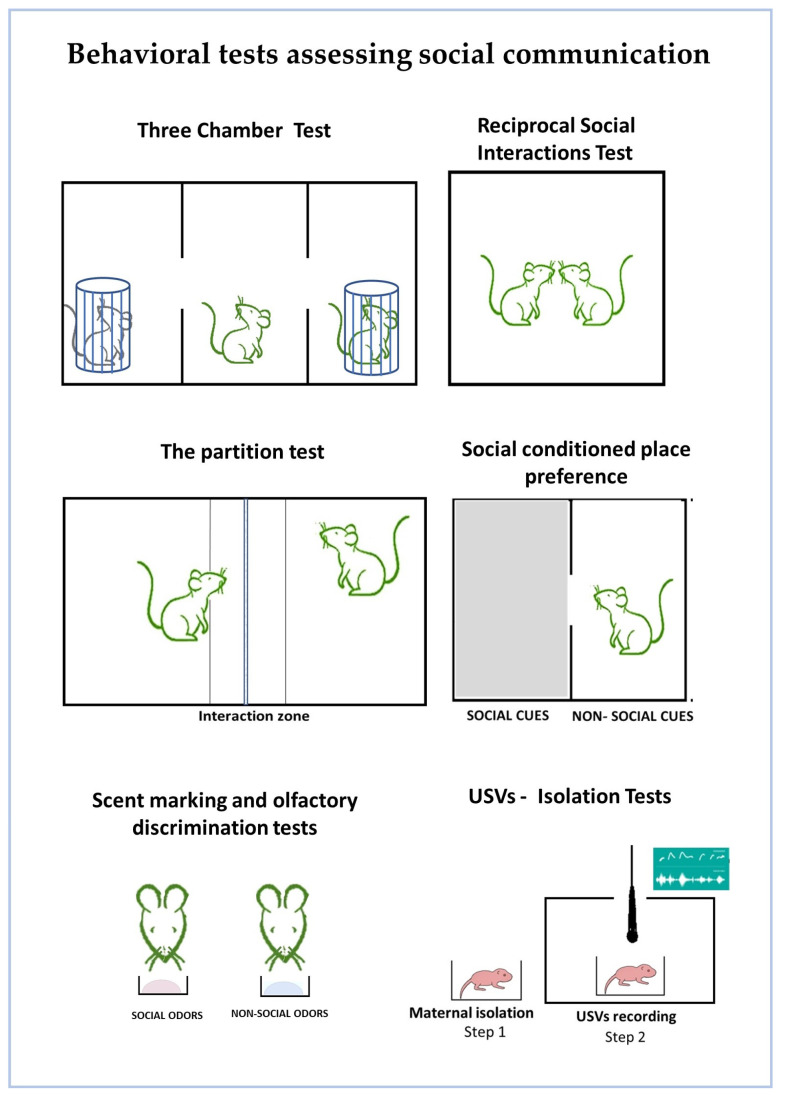



#### 3.1.2. The Partition Test for Abnormalities in Social Behavior (Figure 1)

The partition test, much like the 3-chamber test, is employed to assess social behavioral abnormality in rodents and other animal models of autistic-like behavior [5]. This test is conducted preferably in a standard home cage. Nevertheless, a standard apparatus is routinely used, especially when the system is automated [38]. As the name applies, the cage is partitioned into two equal halves using a perforated wall. The wall can be transparent or opaque but allows for sensory contact between a test animal and a conspecific (stranger). The test involves two phases of habituation and task performance. In the first phase, the test subject is introduced to one part of the partition and is allowed variable time (5 or 10 min) to explore and get familiar with the empty arena. This is followed by the second phase which involves introduction of a conspecific to the other part of the partition, and once again, the test subject is allowed 10 min to explore the conspecific. The time spent by the test animal near the partition in the second phase compared to the opposing end of the arena is used as a measure of the sociability index [35]. A test subject that fails to spend significantly more time near the partition with the stranger is considered to be socially impaired [38,39]. Additionally, attempts by the test animal to put their nose/paws into the perforations or try to gnaw at the partition could be considered an act of nonverbal communication [40,41]. Changing the stranger with other conspecifics could enable the test to assay for novel and social preference in the animal [35,39,42].

#### 3.1.3. Reciprocal Social Interactions/Social Play Behavior (Figure 1)

This assay investigates the social behaviors exhibited by animals in response to interactions with a conspecific, providing insights into their social dynamics and interaction (play), demonstrating potential behavioral anomalies [43,44,45]. The test is more often used in rats for testing social play behavior in NDDs [46]. The apparatus could be a common home cage as the test is best suited for manual observation [47]. In the first step, a test animal is introduced and allowed to habituate to the test arena for 5 min followed by the introduction of a conspecific with which it will be allowed to interact for another 10 min [47]. An observer measures the duration and frequency by which the test animal initiates actions (involving sniffing, following, chasing, pushing past, crawling over/under, pushing under, grooming, fast-paced wrestling, pouncing, spinning, chasing, biting, and boxing) [48,49]. The primary aim is to assess the degree of social engagement, whereby failure to interact more frequently or reciprocate behaviors is indicative of asocial traits akin to NDDs [50,51].

#### 3.1.4. Socially-Conditioned Place Preference and Motivation (Figure 1)

This test is primarily used to evaluate how rewarding a test animal finds a socially conditioned environment [51]. The test set up consists of two chambers with each chamber’s wall differently marked and with one of the chambers housing a social stimulus. Briefly, the test subject is placed in the first chamber that has special markings on the walls that will serve as a cue as well as a social stimulus (like a conspecific or pheromone). In the second phase, the subject is again placed in a new chamber with different wall markings but lacking the social stimuli. In the final stage, the subject is introduced into the arena with the two chambers opened to each other but without the social stimulus in either of the chambers [43]. The initial and final time spent by the animal in both chambers is recorded. The tendency of a subject to spend more time in the nonsocial chamber is a feature of autistic behavior [52]. Animals with autistic-like behavior are reported to spend more time in the chamber that lacks a social stimulus while healthy animals tend to spend more time in the chamber that has a social stimulus [51,53].

### 3.2. Tests for Social Communication Behaviors

#### 3.2.1. Ultrasonic Vocalization (Figure 1)

USVs have emerged in rodents as valuable behavioral biomarkers in neurodevelopmental research. Rodents emit complex USVs ranging from 20 to 110 kHz, fulfilling various social and emotional functions and reflecting their communicative abilities [54]. These ultrasonic vocalizations in mice, observed across diverse social contexts including mother–pup interactions, juvenile play, adult courtship, same-sex social investigations, and territorial encounters, exhibit significant developmental changes throughout the lifespan, mirroring certain aspects of human social and communicative development. This development is associated with thermoregulation and neural maturation and is also influenced by genetic and environmental factors [55,56].

##### Recording and Analysis of USVs

USV recordings require specialized equipment and controlled environmental conditions. Standard recording protocols employ ultrasonic microphones with a 10–180 kHz frequency response, positioned 10–30 cm above the test arena within acoustically isolated chambers to ensure precise sound capture and minimize noise interference. Environmental temperatures are typically maintained between 21 and 34 °C, depending on the protocol, with colder temperatures sometimes used for higher stress induction [18,57]. Circadian rhythm disruptions frequently appear in many neurodevelopmental disorders, and testing for different circadian behavioral changes is recommended [58].

USV analysis offers one of the few reliable means to assess communication behaviors during early postnatal life [59]. The application of machine learning has further improved the accuracy and throughput of USV analysis, enabling a more comprehensive characterization of vocal repertoires [54]. Advancements in automated USV analysis systems, such as those using machine learning algorithms, have significantly improved the accuracy and efficiency of identifying and classifying call parameters like frequency, duration, and amplitude. These parameters may serve as reliable indicators of social and emotional states in rodents, providing a quantifiable measure of their communicative abilities [60,61].

##### USVs in Behavioral Testing Paradigms

Isolation ests: isolation tests are among the most prevalent methods for assessing social and communication deficits. These tests involve separating pups aged 3–14 days from their mothers and littermates typically for a duration of 3 to 10 min to elicit distress calls. The rate of calling as a function of age in healthy pups is typically described as a shallow inverted U-shaped pattern, appearing around day 3, peaking around day 5, and disappearing around day 12. Parameters such as call rate, duration, and frequency are measured, revealing significant differences between ASD models and controls [18,62,63]. This method is sensitive to ASD-related phenotypes and provides insights into the neurodevelopmental impacts of genetic and environmental factors on early communication behaviors [62]. For example, a number of studies have shown a disruption of the typical U-shaped calling rate function in mice with NDD [64,65].

##### Social Interaction Tests

These tests evaluate social communication behaviors by quantifying the USVs emitted during social encounters such as pairings of unfamiliar adult rodents, maternal–offspring interactions, or courtship behaviors. ASD models often exhibit reduced frequency and complexity of these calls, indicating impaired social interaction and distress communication [66]. The USVs produced during these interactions serve as a proxy for social motivation, communication, and recognition. Alterations in the frequency, complexity, and structure of these calls in ASD models often indicate impaired social interactions and emotional processing. For example, male rodents typically emit complex USVs during courtship, while female rodents may respond with specific vocalization patterns. In ASD models, these courtship-related USVs are often reduced in frequency and complexity, reflecting deficits in social behaviors and communicative intent [67].

ASD model rodents, such as the Shank3 mutant and Cntnap2 knockout mice, show significant reductions in the number of USVs during social interactions, correlating with decreased social engagement and motivation [68,69]. Additionally, ASD model rodents often exhibit simpler spectrographic features, such as fewer modulations in pitch or shorter durations, suggesting diminished communicative flexibility [68]. These alterations in USV patterns suggest impaired social motivation, communication, and emotional processing in ASD models.

##### USV Changes in ASD Pups Following Separation from Their Dam and Nest

Rodent pups emit USVs when isolated from their mothers, serving as critical signals for maternal retrieval and social bonding. Alterations in the quantity, frequency, duration, and timing of these vocalizations are regulated by genetic and environmental factors resulting in the impaired communication development associated with ASD [62,63,64,67]. For instance, BTBR T+tf/J mice, an inbred strain used as a model for idiopathic ASD, exhibit significantly fewer isolation-induced ultrasonic vocalizations (USVs) in early postnatal stages and demonstrate unusual spectral and temporal features in their calls, suggesting impaired social communication development [64]. In addition, the atypical USV sequences, comprising both simple and complex call types, were observed in newborn mice with a Tbx1-heterozygous genetic model associated with ASD. These abnormal vocalizations disrupted effective social communication between pups and their mothers, highlighting the potential of USV analysis as a tool for early detection of ASD-like behaviors in animal models [70]. These alterations in USV profiles during early postnatal development may serve as valuable behavioral biomarkers for ASD-like phenotypes and provide insight into the neurobiological mechanisms underlying social communication deficits.

USVs have been widely used to demonstrate behavioral deviations as well as altered emotional states. A similar distinction of 40 kHz vs. 60 kHz frequency classes has been documented in juvenile or adult mice, with lower frequencies associated with aversive conditions and higher frequencies associated with a large variability in call characteristics and subtypes of calls [71]. Depression-like behavior has been associated with USV changes, including a reduction in the number of typical stress emitted calls, namely flat and one-frequency step-up calls, and an increase in step-down type calls, associated with lower levels of stress and anxiety [72].

##### Common Pitfalls in USV

USV studies in rodents are subject to methodological challenges that can significantly impact data interpretation. Subtle changes in the experimental setup, including the ambient temperature and humidity, and inconsistent recording environments, as well as variations in social context and experimenter-induced stress or lack of experience may result in significantly altered vocal patterns [64,66,67]. Technical limitations in recording equipment may also play a critical role in USV studies. Inadequate microphone sensitivity or improper positioning can lead to missed or distorted vocalizations, thereby skewing data. Recent advancements suggest that combining data from multiple high-quality microphones can enhance the accuracy of USV detection and localization [73]. Data analysis and classification may present another set of challenges. Traditional methods often rely on manual identification and classification of calls, which is time-consuming and subject to human error [73].

Species and strain variability: mice emit USVs that may express their unique arousal and emotional states. Rodent species and even strains within a species can exhibit variability in USV characteristics, and findings in one strain may not be generalizable to others. Age and developmental stage considerations can also influence USV emission. Therefore, age-matched controls and longitudinal studies are essential to accurately assess developmental changes in USV patterns.

When designing studies to test the behavior of rodent pups using USVs after maternal separation, it is crucial to consider the duration of separation, as inconsistent results have been reported due to variations in separation protocols. Short periods of separation may not constitute serious deprivation, while longer separations can induce stress in pups and potentially lead to more pronounced behavioral changes [74]. However, excessively long separations or complete isolation from both mother and littermates may cause severe neural stress, potentially confounding the results and making it difficult to distinguish between ASD-like behaviors and general stress responses [74,75]. Finally, the lack of standardized protocols across research laboratories compounds these issues, making cross-study comparisons challenging and potentially leading to spurious conclusions about rodent behavioral and communicative processes.

To meet these challenges, it is advised to employ advanced analytical tools and machine learning modalities when possible and to provide thorough training for experimenters to reduce variability and enhance reproducibility.

#### 3.2.2. Scent Marking Test (Figure 1)

The scent marking test aims to investigate communicability in rodents. Naturally, rodents communicate via olfactory signals such as scent markings and pheromones [48]. During the experiment, the time spent by the test animal sniffing the scent marking and the number of times it deposits urine near the scent marks is a measure of willingness to communicate. Control and wild type rodents demonstrate significant communicativeness compared to models of an ASD-like phenotype [76,77,78]. The paradigm can be modified for habituation and dishabituation, whereby the animal is habituated with a social odor (pheromone) in the first stage and a nonsocial odor (rose or vanilla) in the second stage. In the final stage, the animal is presented with both odors (social and nonsocial) and the time spent sniffing either of the odors is measured. Socially defective models are reported to show less/no preference for the social odor while normal/wildtype rodents preferred the social odor [51].

#### 3.2.3. Pitfalls to Social Behavioral Tests

The 3-chamber test has no major setback in the design, principle, and principal goal of its application. However, considering the significant rise in conflicting results obtained from social behavioral tests among researchers using this paradigm, there is a need to standardize the protocol and the apparatus to facilitate reproducibility and reliability of outcome. The partition test has challenges relating to the apparatus and interpretation of the outcome. The test animal could spend more time close to the partition without the intention of interacting with the conspecific but rather because the partition spans the entire width of the arena relative to the small size of the rodent. In order to overcome this challenge, the number of intentional approaches towards the partition should be a parameter of the test. The assumption that olfaction is the primary means of communication is accurate. However, findings have shown that the opacity of the partition can reduce the socialization index of rodents by 30% [40]. One of the major pitfalls of dyadic social interaction is that the presence of an observer in the room can influence the test performance. The complex nature of the parameters observed in the test session make automation a near impossibility leaving the test in the hand of a human observer prone to bias/subjective scoring and errors. It is possible to introduce video cameras into the recording room to eliminate the presence of an observer. Although the introduction of video recording would solve the problem of observer presence, it cannot prevent errors from manual judgment of the task. The latter is true on realization that rodent encounters are problematic especially in all-male settings, as fights, aggression, and anxiety often ensue to complicate the definition of test parameters [45]. While there are several methods to accomplish the objectives of the scent marking test, olfactory habituation/dishabituation (OHDH) is demonstrated to be the best paradigm of choice because of its ability to discriminate a social deficit from an olfactory deficit [43]. In the case of other approaches to the test, it is recommended that Pup’s nest seeking or olfactory discrimination test should be performed early in development to dismiss doubts of olfactory impairment that might arise in the future. Recommendations for a scoring method are plausible, considering that a social behavioral deficit is a shared attribute among NDDs including ASD [5,28,29], ADHD [28,31], OCD [35], and SZ [79,80,81].

In studying social behaviors, it is crucial to employ multiple corroborating tests to effectively identify social impairments and ensure a comprehensive evaluation across various behavioral domains, including social, motor, cognitive, and anxiety-like behaviors. This multifaceted approach is necessary to reveal unconfounded NDDs such as an ASD-like phenotype, as no single task can adequately measure social behavior or communication. Relying on too few tests may lead to artifactual results, thereby masking the significance of potential confounds in behavioral assessments.

### 3.3. Test Paradigms for Restricted Repetitive Behavior

#### 3.3.1. Marble Burying (Figure 2)

To examine rodents for the presence of obsessive/compulsive behavior, typical in ASD, ADHD, anxiety, and OCD, the marble burying test is frequently employed [47,82]. Repetitive behaviors in rodents often manifest as excessive digging behavior when presented with glass balls in a novel arena [83]. Repetitive and compulsive-like digging behavior is measured by counting the number of balls a test animal buries during a test session [84,85]. The experimental set up typically consists of 20 glass/marble balls in a standard cage with clean bedding materials. Clean cages are filled up to 4–5 cm with bedding material, and the balls are gently placed in four columns by five rows on top of the bedding. The test rodent is then introduced to the arena from one corner to prevent the balls from scattering. The test duration is normally 30 min, during which the rodent is left alone without interference to interact with the balls. At the end of test, the number of marbles that the subject buried by 2/3 of their diameter is used to measure the repetitive digging behavior [36]. NDD models, especially autistic-like models, have a high tendency to bury more balls than wildtype and control animals [86,87].
Figure 2Behavioral tests for assessing restricted repetitive movements. The diagrams present a graphical illustration of the mostly used behavioral tests for the evaluation of restricted repetitive movements: self-grooming; marble burying; and Barnes reversal tests.
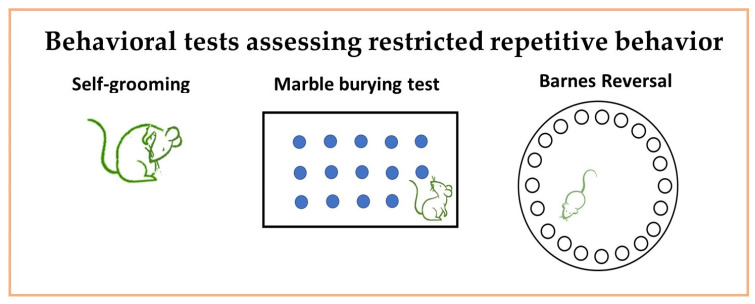



#### 3.3.2. Barnes Reversal (Figure 2)

Reversal of the Barnes maze paradigm is used to examine behavioral inflexibility and spatial memory in rodent models of NDDs such as ASD [5,43]. The apparatus consists of a circular platform (92 cm in diameter) with 20 holes of 5 cm diameter each equally spaced at the perimeter. One of the holes leads to a dark escape box that protects the test subject from the brightly illuminated test room. In experiments assessing repetitive behavior, the protocol consists of five phases: (1) adaptation; (2) forward acquisition training; (3) forward test trials; (4) reversal training; and (5) reversal test trials. During adaptation, the test subject is placed in the middle of the platform and covered for 10 s with a dark box. This is followed by the 2nd phase during which the subject is uncovered and trained to learn the location of the hole housing the escape box within 3 min. After learning the position of the escape hole, the 3rd phase is when the subject is introduced and allowed 90 s to find the location of the hole during which its performance is scored. Investigators can measure the latency to find the correct hole and the number of incorrect holes explored (hole poking error and the length of the exploration path) [38,88,89]. In the reversal phases, the experiment is basically repeated in later days with the position of the escape hole changed by 180 degrees. The training and test are repeated with the escape position reversed, and once again, the scores are measured [90,91]. The maze could be divided into four quadrants so that the subject’s duration of time spent in the escape hole quadrant is measured as well [92].

#### 3.3.3. Repetitive Stereotype Behaviors (Figure 2)

Repetitive behaviors in rodents can be observed in normal activities like self-grooming and bedding [90]. In rodents exhibiting ASD or OCD like behavior, these activities continue for an unusually longer time than normal self-grooming [93,94]. Stereotypic behaviors like backflipping, circling, and jumping are performed more frequently in animals exhibiting autistic-like behavior compared to healthy ones [44]. Following habituation, a subject is observed and scored for cumulative time spent grooming, digging, or chewing. Time spent exhibiting a particular action between models of NDDs and normal subjects are compared as an index of impaired repetitive behavior [94].

##### Pitfalls to Restricted Repetitive Behavioral Test

The marble burying test has some challenges facing the validity of results obtained from the studies mainly due to the multipurpose application of the test outcome. Sometimes excessive digging is related to an anxiety disorder and in other cases is judged as an OCD symptom [95]. Most recently, the test has gained popularity and acceptance among ASD researchers as an important tool for studying repetitiveness in rodent models of ASD-like phenotypes [5,36,96,97].

Questions have been continually raised regarding the validity and reliability of marble burying test outcomes. One of the few problems is the lack of standardized arena dimensions which stem from the fact that the test is classically performed using a home cage. Since different researchers use varying cage sizes, the arena size affects the number of balls and the proximity of balls to each other. Because of this, there are balls that are often covered by bedding material from digging adjacent balls. The nature of bedding material also matters, as saw dust and wood shaving spread further from digging than other types of bedding materials. Some investigators allow the task to proceed with the cage open (with some animals leaving the cage and being returned) while others cover the cage with a lid to overcome the eventual escape of the mice that occurs in many of the cases. The Barnes test is primarily used to assess spatial memory in rodent models of cognitive impairment. However, the introduction of the reversal aspect empowers the paradigm to simultaneously assess both spatial memory and repetitiveness in rodent models of NDDs and other psychiatric diseases [98]. The Barnes maze, although lacking spatial continuity, has the advantage of a dry platform which removes the stress of water and forced swimming that are the major pitfalls for the Morris water maze (MWM) [89,91]. Impaired memory has been reported as one of the comorbidities of ASD and other neurodevelopmental and neuropsychiatric disorders [43,88]. Therefore, the Barnes maze paradigm is considered very advantageous, not only for the less stressful environment it offers but also for the benefit of studying memory together with repetitiveness. The presence of an investigator might be a conflicting source of external influence on the animal’s behavioral task performance. It is important for the observer/examiner to minimize movement and meticulously ensure all test subjects and experimental groups received equal measures of all manipulations in each test performance.

### 3.4. Tests for Cognitive Impairment

#### 3.4.1. Morris Water Maze (MWM) (Figure 3)

The MWM is a test paradigm for assessing spatial learning and memory and reversal learning in rodent models of NDDs [99]. With the aid of spatial cues, subjects learn the location of a platform hidden underwater as the escape route from a circular pool. The apparatus is a round water pool of about 100 cm in diameter and water with a temperature of 22 °C. The hidden escape platform is placed 1–2 cm below the water in one quadrant of the circular pool, and the water is colored to conceal the platform. Test subjects are trained for five consecutive days in the apparatus to learn the location of the platform by swimming using distinct visual cues placed around the maze [100]. During the training days, mice are introduced into the pool four times per session via four different locations, and the speed and latency to locate the platform is recorded. Each trial ends when the subject escapes to the hidden platform or is placed on it by the experimenter if it could not locate it. On the test day after training days, the test is performed again, this time with the platform removed [101]. The time the subject spent in each quadrant, and the number of times it crosses the original location of the platform, is recorded. The preference ratio can be computed as the time spent in platform quadrant/time spent in the other three quadrants [33,88,100].
Figure 3Behavioral tests for assessing cognitive impairments. The diagrams present a graphical illustration of the Y-maze, Morris water maze, novel object recognition, and fear conditioning that are used for the evaluation of learning and memory impairments.
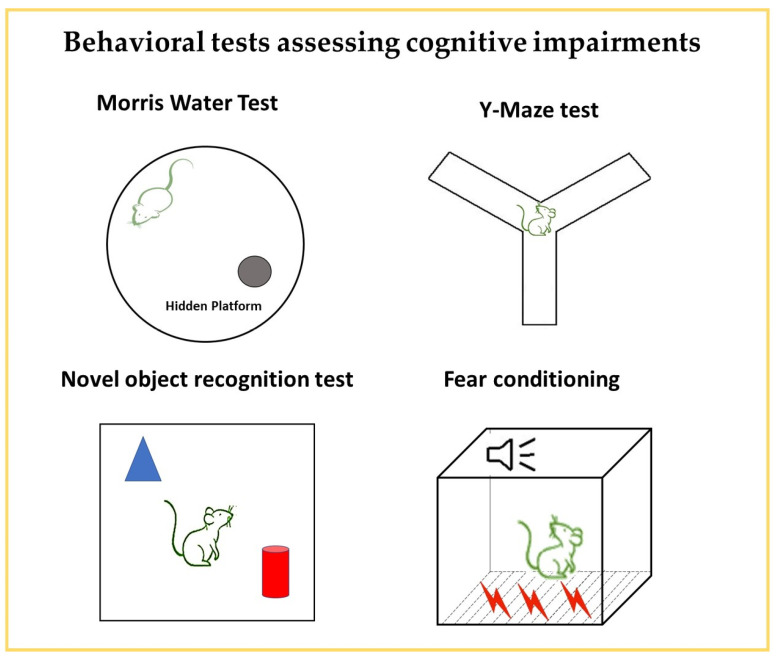



#### 3.4.2. Y-Maze Spontaneous Alternation (Figure 3)

This test is employed with rodents to measure hippocampal function in the context of short-term memory (working memory) [88]. The test arena is a maze that consists of three identical arms (forming a Y shape) each being 40 cm in length and 10 cm in width and height. During the experiment, animals are placed in one of the arms from which they are allowed 5 min or more to sequentially explore other arms freely. Test animals with intact working memory will normally alternate between the arms after each visit by moving unto a new arm and not going back to a previously visited arm. An animal with impaired cognition tends to show reduced spontaneous arm alternation that accompanies increased arm revisit which indicates impairment in memory events [33]. The percentage of alternation is calculated as the ratio of the total number of alternations (triad containing entry into all three arms) to the total number of entries minus 2 multiplied by 100 [102]. Deficits in spontaneous alternations are characteristic of the ADHD and schizophrenia phenotypes and could be ASD-like comorbidity features.

#### 3.4.3. Novel Object Recognition (NOR) (Figure 3)

The NOR is also used to assess intellectual difficulties that may be present in most NDDs and confounds ASD [43]. The test which measures memory is based on rodents’ innate tendency to explore novel objects than a familiar one [88,103]. The apparatus is an open box of about 50  ×  50  ×  40 cm. The test proceeds by first familiarizing the subject with the arena for a given amount of time (5 min). This is followed by introducing two identical objects (including size, color, and shape), and the test subject is allowed to familiarize itself with them [103]. Finally, one of the familiar objects is replaced with a novel object (only the size should be maintained), and the subject is allowed to explore the familiar versus the unfamiliar object. Placing objects in the same position is important to avoid localization-related differences. The time spent exploring novel and familiar objects are recorded, and the discrimination index computed as follows: (time with a novel object—time with a familiar object)/(time with a novel object + time with a familiar object) [104,105].

#### 3.4.4. Fear Conditioning (Figure 3)

The test is used to assess fear processing and to associate an aversive foot shock with contextual or sound cues in rodent models of an autistic-like phenotype. NDD models are reported to display elevated fear memory and generalization for sound and contextual cues [106]. The protocol begins with a habituation phase when subjects are allowed to acclimate to the conditioning chamber without stimuli. After habituation, the conditioning phase involves presenting a sound queue for 30 s followed immediately by an aversive shock, repeated several times with inter-trial intervals, and then the subjects are returned to their home cage. After the conditioning phase, subjects undergo a contextual fear test, where they are placed back in the chamber with the sound cues but without the aversive shock [15]. Since rodents express fear as freezing behavior, the time spent in a freezing state is taken as an index of fear. Data analysis focuses on quantifying freezing behavior during both tests, with appropriate statistical methods employed to evaluate differences between conditioned and control groups [107,108].

##### Pitfalls to Tests for Cognitive Impairments

The Y-maze test is a valuable tool for assessing spatial memory, anxiety, and repetitiveness in rodents, yet it is considered not to be as definitive as the Barnes maze and other more specialized learning/memory tests [90]. Pitfalls in the Y-maze could include variability in individual behavior as some animals may exhibit inherent differences in exploration tendencies or anxiety levels. The novelty of the environment and maze design can influence behavior, potentially confounding results. It is important to allow a familiarization time of 1–2 min before recording subject performance. The reliance on spontaneous alternation may also affect the ability to discern between memory and motivational factors [109]. There are inconsistencies between laboratories in terms of lighting conditions, maze size, and test duration that need to be addressed to lessen the gap in results. Therefore, careful consideration of these variables is essential for robust conclusions in studies utilizing the Y-maze test.

There are a few challenges in relation to the MWM despite it being the most robust and reliable test for assessing spatial learning and memory. The aquatic stress involved can impact rodent performance because of differences in anxiety and swimming ability. This presence of water could lead to extraneous behaviors quite unrelated to spatial learning, particularly in mice [95,97]. Too much focus on escape time in addition to individual subjects’ escape strategy could mask subtle memory impairments. A lack of reporting/regulation of the water temperature and maze size are commonly practiced in many studies despite these variables affecting task performance and result variability. Although considered the cornerstone of the field, careful consideration of these factors is crucial for valid conclusions in MWM studies.

In NOR, variability in the objects explored can affect replicability, as some test subjects may show little interest in the objects, necessitating strict inclusion/exclusion criteria based on performance levels. The transient nature of novelty means that attention to new objects may diminish quickly, which impacts the reliability and validity of results. It is important, therefore, to optimize the test duration as many studies fail to control the time spent with objects during familiarization, potentially skewing retention data and leading to inaccurate conclusions. It is also important to acclimatize the subject with the arena before any object introduction.

The fear conditioning test measures associative and working memory. Contextual and cued conditioning components of the paradigm mainly assess specific associative learning, potentially overlooking other cognitive responses. The specific nature of cues can limit the applicability of results. Moreover, the fear conditioning test has a major pitfall in the reliance on foot-shock as a stressor capable of inducing significant anxiety and distress in the subjects. This raises ethical concerns regarding animal welfare, and, therefore, compliance with animal welfare guidelines and minimizing distress must be considered as paramount. The level of individual subject’s response to stress differs and may introduce freezing behaviors that are not components of memory.

### 3.5. Tests for Sensory Stimuli

#### 3.5.1. Prepulse Inhibition (PPI) (Figure 4)

Animal models of psychiatric disorders exhibit central deficits in sensorimotor gating (i.e., abnormal PPI) of the robust acoustic startle response (ASR), which can be reliably measured [81,103,110]. The apparatus for testing consists of two soundproof test chambers with dimensions of about 40 × 40 × 60 cm, with a restrainer in the chambers, loudspeakers to generate acoustic stimuli, and a sensor to measure the startle response. The test is made up of a habituation day during which the test subject is acclimatized to the test environment. Acclimatization includes exposing subjects to a steady background noise of 70 dB (decibel) and multiple startle stimuli of 120 dB to test hearing and motor functions. On the test day, the subject is exposed to the arena and recorded over several exposures to a weaker acoustic stimulus (prepulse) followed by an intense startle stimulus (pulse). A sudden puff of air can be used as a tactile stimulus to elicit a startle response which can then be quantified [110]. By measuring the startle amplitude at the beginning and end of a session, the percent of prepulse inhibition can be calculated using the formula %PPI = 100 × ([startle-only units − (pre-pulse + startle units)]/(startle-only units)).
Figure 4Behavioral tests for assessing sensory impairments. The diagrams present graphical illustrations of the hot-plate and tail-flick test that are used to assess the nociception threshold of sensory perception, while the prepulse inhibition test is used to evaluate the startle reflex induced by unpleasant sound signals.
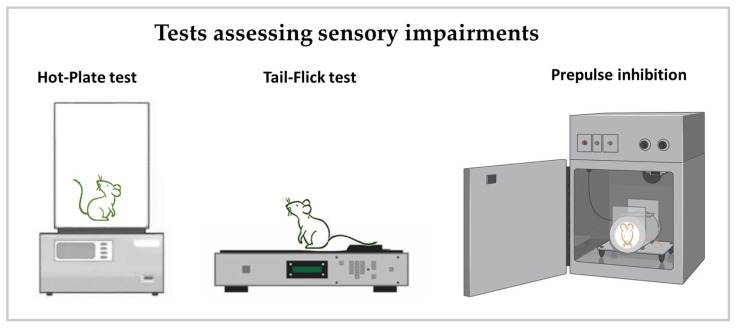



#### 3.5.2. Nociception (Figure 4)

This paradigm is used to assess rodent reactivity to sensory stimuli. To achieve this purpose, the hot-plate and tail-flick tests are frequently employed. The subject is basically placed on a plate that is heated and kept at a constant temperature of about 55 °C. The latency to the first reaction (licking of paw or jumping) of the animal is recorded. Similarly, in the tail-flick test, the subject tail is placed in hot water or exposed to a beam of light, and the latency to flick the tail is recorded [110,111].

### 3.6. Tests for Anxiety-like Behavior

#### 3.6.1. Elevated Plus Maze (EPM) (Figure 5)

The elevated plus maze (EPM) and dark/light box tests are suitable and widely used tests for anxiety [16,111]. These tests explore the approach–avoidance conflict between mice’s innate curiosity to explore novelty and mice’s preference to be in closed/dark environments instead of opened/illuminated places. The apparatus is a plus-shaped maze whereby each arm measures about 30 cm in length and 8 cm in width. Two opposing arms have 17 cm high opaque walls around them except for the central entrance (closed arms); the other two opposing arms are not walled (open arms). This arrangement creates a central platform to which animals are introduced for observation. The maze is pivoted at a height of 70–100 cm above the floor to discourage test subject from jumping off the platform. During the experiment, the subject is placed on the central platform with its face towards the open arm [112]. The subject is then given time to explore the arena for a given time according to the experimental design [100,113].
Figure 5Behavioral tests for assessing locomotion and anxiety. The diagrams present graphical illustrations of the open field test which is mainly used to assess locomotion, motivation and, anxiety levels, whereas the elevated plus maze and dark/light box tests are used to assess anxiety.
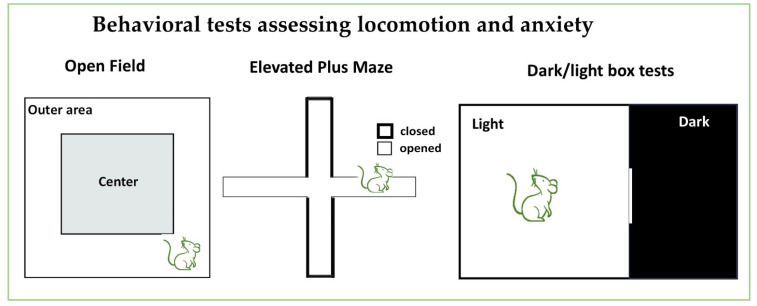



#### 3.6.2. Dark/Light Box Tests (Figure 5)

The dark/light apparatus is made up of a white-painted and brightly illuminated open box that is connected to another black-painted, non-illuminated closed box. The boxes are rectangular and vary in their dimensions from lab to lab. During behavioral observation, subjects are placed gently in the middle of the white box with their faces directed opposite to the connecting opening [15,114]. The frequency of entries and the total time spent in the open/lighted, the closed/dark region, and the central zone is recorded and analyzed [115]. The increased time spent in open arms of the EPM or in the illuminated part of dark/light box is an indication of a lower degree of “anxiety” in the animal models of autistic-like behavior. 

#### 3.6.3. Open Field Locomotory (OF) Test (Figure 5)

This test is widely employed for the assessment of general locomotion, exploration, and anxiety behavior. The paradigm consists of a square arena that is divided into nine equal parts [100,112]. For the purpose of examination, a subject is introduced to the center of arena and allowed 5 to 10 min to explore the open field, and its activities are observed and recorded. Locomotory/exploratory behaviors like horizontal and vertical activity, movement and rest time, and total distance travelled are measured [80,115]. The latency, duration, and the number of visits spent in the center of the arena or on the wall are recorded for computing the anxiety level [15]. In addition, repetitive and stereotyped behaviors of the subject can be determined by observing the self-grooming, spinning, jumping, and rearing behavior [113]. Rodent models of NDDs especially ASD-like and ADHD-like models are reported to spend more time grooming in an abnormal fashion and spend more time near the walls than the center of the arena [32,86,93]

#### 3.6.4. Pitfalls to Anxiety Tests

For studies that involve behavioral tests for anxiety-like behaviors including the EPM, the dark/light Box, and the OF tests, it is advised to measure and report that lighting conditions are optimal and consistent to ensure that animal responses align with expected outcomes. Consistent reporting of the species, age, and sex of subjects is crucial for understanding these behaviors. Meta data analysis showed that studies have reported durations as low as 4 min to as high as 30 min, pointing to the lack of standardized optimal latency and duration in the application of these tests [116]. Most studies use test durations of 5 to 10 min; however, novelty in a new environment can induce neophobia, skewing results toward increased anxiety due to insufficient habituation. Conversely, exceeding 10 min may reduce anxiety responses as animals acclimate. Therefore, a 10 min duration is recommended, with the first 5 min used for assessing anxiogenic responses and the last 5 min for evaluating anxiolytic responses.

In the elevated plus maze, lighting and noise can significantly influence performance and alter anxiety responses, as the test primarily measures approach–avoidance behavior, which may not fully capture the complexity of anxiety disorders. Similarly, in the dark/light Box, rodents’ preference for dark environments can confound interpretations, and environmental factors like light intensity and arena size may also affect results. The open field test may not distinguish between anxiety and general activity levels, as increased movement can arise from curiosity or fear. Thus, careful consideration of these factors is essential for valid conclusions in anxiety-related studies.

### 3.7. Tests for Motor Coordination and Balance

#### 3.7.1. Rotarod Paradigm (Figure 6)

The rotarod test is a conventional assessment tool for motor coordination in rodent-based experimental studies on NDDs and other brain disorders that involve impairment in motor coordination [43,88]. Using a rotating rod paradigm, motor coordination and balancing as well as motor skill learning can be tested by recording the amount of time a subject remains moving on the rod while an improved performance across repeated test sessions is a marker of motor learning ability [117]. The rotarod apparatus consists of a rotating rod that accelerates in predetermined time intervals. The floor is padded to prevent injury from falling off because the rod lies horizontally at a height that is high enough to scare subjects from jumping [118]. The latency to fall over at varying speeds are recorded as indications of the motor coordination function and similar judgement is applicable to the beam walking (static rod) test for motor function in rodents [119].
Figure 6Behavioral tests for assessing balance and motor coordination. The diagrams present graphical illustrations of the rotarod and the beam walking tests that are useful for evaluation of motor coordination and balance.
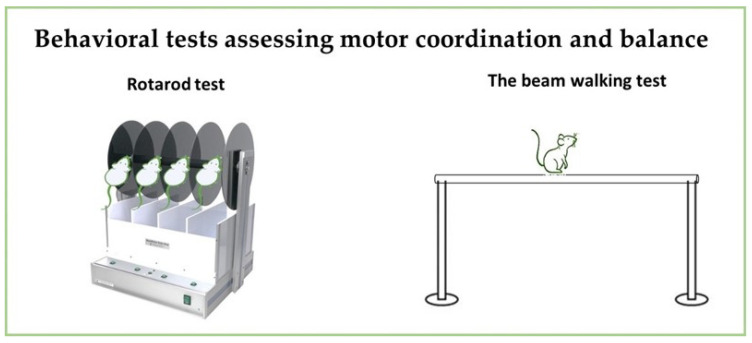



#### 3.7.2. The Beam Walking Test (Figure 6)

This is a widely used alternative paradigm to the rotarod test, suggestive of its better sensitivity than the rotarod test in detecting deficits in motor coordination because, in addition to latency, the test uses quantitative values such as walking distance and number of slips to accomplish the task on the narrow beam [120,121]. The beam walk apparatus is a wooden beam of 80 cm of length and 1 cm of width at a height of 60 cm. At one end of the bar, a dark escape box is attached 10 cm from the finish line, while a line on the free opposing end marks the starting point. During training and test days, the test subject is placed on the elevated beam the free starting end and the latency to cross the beam and the number of foot slips are recorded as measures of incoordination in NDD rodent models [122].

The beam walking paradigm is not popular compared to the rotarod test. However, it appears to have the additional advantage of assessing balancing and sensorimotor gating. It also has a low-cost advantage and is a simple paradigm.

To further probe the deficits assessed in the rotarod and beam walking tests, other tests of coordination like footprint analysis of locomotor gait, mesh climbing, swimming, pole climbing, and the staircase test can be employed.

#### 3.7.3. Pitfalls to Tests for Coordination and Balance

The rotarod test, even though widely used for assessing motor performance in mice, has several notable limitations that can complicate the reproducibility and interpretation of results. One such issue is that, while some animals may cling to the beam and rotate alongside the rod instead of properly balancing on the rotating rod, others may fall immediately upon being placed on it. Additionally, the apparatus may not accelerate sufficiently and uniformly to accurately detect motor incoordination effectively, and if the height is not optimally raised, animals can be encouraged to jump off the rod. Variability in test results often also arise from different test paradigms and apparatus designs (including the lighting condition, diameter and texture of the rod, and width of the compartments), which can yield inconsistent outcomes under similar conditions. Furthermore, certain strains of mice are known to perform poorly on the rotarod, raising questions about the validity of conclusions drawn from such tests. Factors like stress, learning effects from repeated trials, and test environment can further influence results. It is crucial to standardize training protocols and consider the mentioned factors to ensure reliable assessments of motor coordination.

As animals often jump off during task performance, it is important to ensure that the beam is elevated high enough. Lack of motivation can be a confounding factor if the subjects are subjected to multiple training sessions that can translate to uncooperativeness. The surface of the beam should have a uniform smoothness without any textures or protrusions that subjects might grip. Animals from different test groups should be of the same age, weight, and strain. Weight differences might be an important factor in this test paradigm; considering that older and heavier animals tend to be slow and are likely to have more slips on a beam of equal width. Finally, for the purpose of assessing balancing in rodents, it is important to consider the width of the beam because abeam that is too wide may not be necessary or effectively assess balancing.

## 4. Discussion

We have presented a variety of methods assessing different behavioral traits in rodents that are used to measure behaviors which characterize neurodevelopmental and psychiatric disorders—NDDs. For each set of behavioral tests, we described the possible pitfalls, stressing the likelihood of errors that might affect the results. While deviations in many of these behaviors are found in ASD, most of them are not specific to ASD-like behavior as they may also characterize other neuropsychiatric disorders. Hence, in many NDDs, the existence of a specific combination of such deviation points to the diagnosis of a specific disorder. However, for a precise diagnosis, a minimal number of typical behaviors exhibiting a significant severity must be present.

The DSM 5 and other diagnostic measures require grading the severity of the observed behavioral deviations for proper diagnosis. Indeed, the behavioral assessment tools used for diagnosis generally also grade the severity of the deviations. This is true, for example, with the common diagnostic tools for ASD such as the Child Autism Rating Scale (CARS), Autism Diagnostic Observation Schedule (ADOS), Autism Diagnostic Interview-Revised (ADI-R) scale, and other diagnostic tools [1].

The different tests used for the assessment of behavioral changes in animals, especially in rodents, are generally adequate to assess all specific and typical autistic-like behaviors. However, studies assessing only a few behavioral tests for the demonstration of a specific NDD (i.e., only tests for social interaction and communication or repetitive behaviors and anxiety to demonstrate ASD-like behavior) may be insufficient. Such studies can be used, however, to define specific traits of autistic-like behavior, i.e., communication difficulties, repetitive behavior, restricted interests, abnormal response to sensory stimuli, and others, but not the complete diagnostic set. Hence, an accepted scoring system or, at least, a definition of the severity, similar to that in humans, seems to be important for all animal models that mimic human neurobehavioral and neuropsychiatric diseases.

In the last years, there have been several attempts to standardize behavioral data and provide relative severity to behavioral tests and gender differences.

### Integrated Behavioral z-Scoring

In the research of NDDs, z-score can be used to standardize complex pathophysiology into a single paradigm, similar to diagnosing mental health conditions in humans. z-scores convert diverse measurements to a standard scale by subtracting the population average from an individual raw value and dividing by the population standard deviation, allowing researchers to compare and interpret data across different studies and variables [123,124]. Thus, Guilloux et al. [123] used this technique to normalize multiple anxiety- and depressive-like behavioral tests of paradigms into a single emotionality z-score index. The authors emphasize the significance of the *z*-normalization approach as an analysis that increased the sensitivity and reliability of behavioral phenotyping in mice [123].

Kraeuter et al. [124] described the methodology of integrated behavioral z-scoring in rodent studies. Integrated behavioral z-scores can be applied to the psychiatric or neurodevelopmental disorders, such as ASD, and can even be combined with biochemical and molecular change z-scores, forming a comprehensive disease-specific integrated z-score.

Labots et al. conducted a study using the modified Hole Board (mHB) test to evaluate behavioral characteristics in three inbred mouse strains: A/J, BALB/cJ, and C57BL/6J [125]. To analyze the data, they employed z-score calculations using the pooled data from all groups as a reference. This approach was needed as the experimental design lacked a common control group due to the comparison of both sex and strain differences. Labots et al. noted that traditional z-normalization methods would be impossible if the control group had a standard deviation of zero, highlighting the advantage of their pooled data approach [125].

El-Kordi et al. utilized z-score normalization to analyze ASD-like behavior in Nlgn4 null mutants, with higher z-scores indicating greater symptom severity [126]. For males, the ASD severity score encompassed seven behavioral categories spanning the three core symptoms: qualitative impairments in social interaction (social approach behavior, nest building, and aggression), communication deficits (ultrasonic vocalizations during male–female interactions), and restricted, repetitive, and stereotyped behaviors (marble burying and circling behavior) [126]. Similar behavioral categories were identified as relevant for females [127]. Applying the ASD severity score demonstrated near-perfect genotype classification for individual male mice (almost 100% accuracy), while accuracy for female mice was slightly lower, at approximately 80% for correct group assignments. These findings highlight the utility of z-score normalization as a robust method for quantifying and distinguishing ASD-like behaviors across genotypes.

In our previous studies on ASD-like behavioral changes induced by early postnatal administration of VPA, we developed a composite scoring system to classify these behavioral alterations [88]. We integrated z-scores from self-grooming frequency in the open field test, incorrect turning percentages in the T-maze, and social novelty preference in the three-chamber social interaction test as three key behavioral measures. Our findings showed that the ASD composite score was significantly higher in VPA-treated male and female mice compared to controls and was normalized by the co-administration of a methyl donor, S-Adenosylmethionine (SAMe), and VPA. Despite these findings, the study did not address the severity of individual behavioral changes, which could be assessed by quantifying the degree of deviation from control values for each animal.

In clinical practice it is advised to use a four-scale scoring system in the definition of the severity of a NDD: normal, slightly abnormal, moderately abnormal, and severely abnormal [1,5]. The possible scores for each behavioral characteristic are from 0 to 3. Zero being normal behavior and 3 being severely abnormal. As different NDDs are characterized by abnormalities in several specific behavioral traits, a composite z-score for all the behavioral characteristics studied is generally inadequate. Hence, it is expected that each one of the typical behaviors tested will have its own severity score, and then the composite score can be calculated, as in clinical medicine. Moreover, it is expected that for each NDD, most, if not all of the behaviors expected to be abnormal, will be tested. For example, to define an appropriate animal model for autistic-like behavior, it is imperative to assess sufficient behavioral traits that characterize ASD-like behavior: at least two of the three specific social characteristics (social behaviors, social interaction, and communication) and two of the following: restricted and repetitive behaviors, cognitive impairment, anxiety-like behavior, and motor coordination. In addition, there is a need for individual scoring of each of the behaviors tested.

As there are differences in the clinical presentations of some NDDs between males and females, it seems important to carry out behavioral studies on both sexes.

## 5. Conclusions

We have described a variety of behavioral tests for mice that are commonly used to describe animal models of human NDDs. We pointed out the possibilities of serious pitfalls in the performance and interpretation of these tests. Investigators should therefore be aware of these possible drawbacks and consider them carefully before carrying out neurobehavioral studies, as they should be reflected in the assessments of the results. We also pointed out the need to use sufficient and aimed behavioral tests to define a model of a specific NDD. Due to gender differences in many human NDDs, it is appropriate to study males and females. To mimic the clinical definitions, it is important to assess the severity of each specific behavioral change.

It is recommended that behavioral experts decide on these salient factors which may affect the progress of preclinical findings. Consensus can be reached on the various issues raised by us, reducing the possible difficulties interfering with the translation of animal behavioral deviations to human disorders.

## Data Availability

No new data were created or analyzed in this study.

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
