# Peer review of "Methods for Assessing Neurodevelopmental Disorders in Mice: A Critical Review of Behavioral Tests and Methodological Considerations Searching to Improve Reliability"

_neurosci, 2025, doi:10.3390/neurosci6020027_

Round 1
Reviewer 1 Report
Comments and Suggestions for Authors
The manuscript presents a valuable discussion on neuromethods for assessing neurodevelopmental disorders (NDDs) in rodent models. However, several aspects require further elaboration and refinement to enhance the clarity, novelty, and scientific rigor of the review.
Comment 1: The discussion of potential pitfalls in the described methodologies is relatively limited, and little attention is given to strategies for mitigating them. Addressing these aspects in greater depth would significantly improve the manuscript's utility for researchers in the field.
Comment 2: In “3.0. Behavioral tests assessing specific behavioral traits” It would be beneficial to explicitly differentiate between protocols for rats and mice within each section describing behavioral tests. Moreover, including an introductory paragraph discussing the advantages and limitations of each species in specific paradigms would provide a more comprehensive evaluation of their use in NDD models. For instance, rats are more commonly employed in social behavior tests due to their inherently higher sociability compared to mice, which tend to be more aggressive depending on the strain. If a comparative approach is not feasible, the authors may consider focusing exclusively on mouse models to maintain consistency.
Comment 3: The organization of the review could be improved to enhance readability and logical flow, particularly for readers “outside the field”. Specifically, for each behavioral test described, a dedicated subsection addressing its specific pitfalls should be considered, instead of discussing all the pitfalls at the end of the paragraph. This would facilitate a clearer and more structured presentation.
Comment 4: “2.1 Litter Effect”: The litter effect is crucial for the reliability and reproducibility of behavioral tests, as rightly acknowledged by the authors. However, the manuscript does not adequately define this concept, nor does it provide a discussion of methodological approaches to control or minimize it. Given its well-documented influence on experimental outcomes in both NDD models and their control groups, a more explicit consideration of this issue is warranted. Established strategies for addressing the litter effect, including for example, selecting only one animal per litter at random, using multiple animals per litter and averaging their values, or applying a mixed-effects model for statistical analysis when using multiple animals per litter should be considered. Incorporating a discussion of these approaches would strengthen the methodological rigor of the review.
Comment 5: 2.2 Uniformity of Pregnancy Timing: In line 177, it is unclear whether the authors are referring to the timing of mating or that of testing. To avoid ambiguity, this aspect should be explicitly clarified, and if necessary, the sentence should be revised for improved precision.
Comment 6: 3.1.3 Reciprocal Social Interactions: While the discussion of reciprocal social behaviors provides valuable insights, it would be beneficial to also mention the role of social play behavior in rodents, particularly in rat models of ASD. Social play, including rough-and-tumble interactions, is an essential component for assessing social functioning and detecting social impairments in these models. Research has demonstrated that alterations in social play behavior are evident in various animal models of ASD and are associated with the core social deficits observed in these disorders. Including an evaluation of social play behaviors alongside reciprocal interactions would provide a more comprehensive assessment of social dynamics in NDD models and offer an additional measure for identifying social impairments.
Author Response
We thank the reviewer for the valuable comments
Comments and Suggestions for Authors
The manuscript presents a valuable discussion on neuromethods for assessing neurodevelopmental disorders (NDDs) in rodent models. However, several aspects require further elaboration and refinement to enhance the clarity, novelty, and scientific rigor of the review.
Comment 1: The discussion of potential pitfalls in the described methodologies is relatively limited, and little attention is given to strategies for mitigating them. Addressing these aspects in greater depth would significantly improve the manuscript's utility for researchers in the field.
We added some data regarding attempts to reduce the impact of possible pitfalls, especially in the section on general pitfalls (section 2) that was changed according to the important reviewers suggestions
Comment 2: In “3.0. Behavioral tests assessing specific behavioral traits” It would be beneficial to explicitly differentiate between protocols for rats and mice within each section describing behavioral tests. Moreover, including an introductory paragraph discussing the advantages and limitations of each species in specific paradigms would provide a more comprehensive evaluation of their use in NDD models. For instance, rats are more commonly employed in social behavior tests due to their inherently higher sociability compared to mice, which tend to be more aggressive depending on the strain. If a comparative approach is not feasible, the authors may consider focusing exclusively on mouse models to maintain consistency
Since most studies we discuss were done on mice, we prefered to omit the data related to rats and concentrate only on mice. We therefore made the relevant changes, including the change in the title.
Comment 3: The organization of the review could be improved to enhance readability and logical flow, particularly for readers “outside the field”. Specifically, for each behavioral test described, a dedicated subsection addressing its specific pitfalls should be considered, instead of discussing all the pitfalls at the end of the paragraph. This would facilitate a clearer and more structured presentation.
We believe that this might be a good idea, but it will necessitate repetitions of similar pitfalls many times as we generally describe several tests for the same measured behavior, which have similar pitfalls. We therefore prefer to keep all pitfalls at the end of the description of all tests measuring a specific behavior.
Comment 4:“2.1 Litter Effect”: The litter effect is crucial for the reliability and reproducibility of behavioral tests, as rightly acknowledged by the authors. However, the manuscript does not adequately define this concept, nor does it provide a discussion of methodological approaches to control or minimize it. Given its well-documented influence on experimental outcomes in both NDD models and their control groups, a more explicit consideration of this issue is warranted. Established strategies for addressing the litter effect, including for example, selecting only one animal per litter at random, using multiple animals per litter and averaging their values, or applying a mixed-effects model for statistical analysis when using multiple animals per litter should be considered. Incorporating a discussion of these approaches would strengthen the methodological rigor of the review.
Thanks for this comment. We expanded the discussion on litter effects as suggested
Comment 5: 2.2 Uniformity of Pregnancy Timing: In line 177, it is unclear whether the authors are referring to the timing of mating or that of testing. To avoid ambiguity, this aspect should be explicitly clarified, and if necessary, the sentence should be revised for improved precision.
We made this change and several other changes expanding this paragraph
Comment 6: 3.1.3 Reciprocal Social Interactions: While the discussion of reciprocal social behaviors provides valuable insights, it would be beneficial to also mention the role of social play behavior in rodents, particularly in rat models of ASD. Social play, including rough-and-tumble interactions, is an essential component for assessing social functioning and detecting social impairments in these models. Research has demonstrated that alterations in social play behavior are evident in various animal models of ASD and are associated with the core social deficits observed in these disorders. Including an evaluation of social play behaviors alongside reciprocal interactions would provide a more comprehensive assessment of social dynamics in NDD models and offer an additional measure for identifying social impairments.
Basically, reciprocal social interaction and social play behavior are similar. Since we omitted the studies on rats, we added a sentence related to rats but no more than that: "The test is more often used in rats for testing social play behavior in NDD " [Servadio M et al, Trasl. Psych. 2016, PMID 27676443, DOI 10.1038/tp.2016.182
Reviewer 2 Report
Comments and Suggestions for Authors
It is an important review article for animal behavioral researchers to understand the challenges and measures to be taken for improving their data. However, a few things need to be addressed such as,
-> While authors have given explanations for each and behavioral experimental methods, it doesn’t look satisfactory from a readers’ s point of view. As such, this review would benefit from developing additional display items, including representational images or diagrams that visually display each behavioral equipment mentioned under “Tests for behaviors” along with its appropriate descriptions.
-> Even though, the authors mentioned different behavioral experiments for each disease conditions and its limitations from other literatures, their objective or purpose of the review needs to be clearly specified in a separate paragraph in the end of the introduction and conclusion as well. What’s new from this article for the readers is the key for a review article.
-> To keep the uniformity either use ASD or autism throughout the manuscript. Using ASD throughout is recommended.
-> In page 18, line 825 , expand or describe “SAMe”.
Author Response
Comments and Suggestions for Authors
It is an important review article for animal behavioral researchers to understand the challenges and measures to be taken for improving their data. However, a few things need to be addressed such as,
-> While authors have given explanations for each and behavioral experimental methods, it doesn’t look satisfactory from a readers’ s point of view. As such, this review would benefit from developing additional display items, including representational images or diagrams that visually display each behavioral equipment mentioned under “Tests for behaviors” along with its appropriate descriptions.
We thank the reviewer for his important comments. The changes are in the text in color
We added six figures, each one showing schematic images of the testing apparatuses used for a specific behavioral trait tested.
-> Even though, the authors mentioned different behavioral experiments for each disease conditions and its limitations from other literatures, their objective or purpose of the review needs to be clearly specified in a separate paragraph in the end of the introduction and conclusion as well. What’s new from this article for the readers is the key for a review article.
Many thanks for this comment. We added a paragraph at the end of the introduction just before the very last paragraph. In color
-> To keep the uniformity either use ASD or autism throughout the manuscript. Using ASD throughout is recommended.
We corrected and changed in all places to ASD
-> In page 18, line 825 , expand or describe “SAMe”.
Done. S-adenosyl methionine is SAMe. This is a methyl donor produced by all living organisms that enhances methylation of DNA, causing silencing of genes.
Round 2
Reviewer 1 Report
Comments and Suggestions for Authors
I appreciate the author's efforts in revising the manuscript. They have adequately addressed all my comments and made the suggested structural improvements. When they could not implement a suggestion, they responded to my concerns point by point. As a result, the paper is now clearer and better organized. I have no further concerns and believe it is suitable for acceptance.
Reviewer 2 Report
Comments and Suggestions for Authors
The authors made appropriate additions and changes during the revision and thus improving the quality of the manuscript significantly.